# Intake of dietary fats and fatty acids and the incidence of type 2 diabetes: A systematic review and dose-response meta-analysis of prospective observational studies

Manuela Neuenschwander[1,2], Janett Barbaresko[1], Claudia R. Pischke[3], Nadine Iser[1], Julia Beckhaus[1], Lukas Schwingshackl[4], Sabrina Schlesinger[1,2]*

**1** Institute for Biometrics and Epidemiology, German Diabetes Center, Leibniz Center for Diabetes Research at Heinrich Heine University Düsseldorf, Düsseldorf, Germany, **2** German Center for Diabetes Research (DZD), Munich-Neuherberg, Germany, **3** Institute of Medical Sociology, Centre for Health and Society, Medical Faculty, Heinrich Heine University Düsseldorf, Düsseldorf, Germany, **4** Institute for Evidence in Medicine, Faculty of Medicine, University Medical Center Freiburg, Freiburg, Germany

* sabrina.schlesinger@ddz.de

## Abstract

### Background

The role of fat quantity and quality in type 2 diabetes (T2D) prevention is controversial. Thus, this systematic review and meta-analysis aimed to investigate the associations between intake of dietary fat and fatty acids and T2D, and to evaluate the certainty of evidence.

### Methods and findings

We systematically searched PubMed and Web of Science through 28 October 2019 for prospective observational studies in adults on the associations between intake of dietary fat and fatty acids and T2D incidence. The systematic literature search and data extraction were conducted independently by 2 researchers. We conducted linear and nonlinear random-dom effects dose–response meta-analyses, calculated summary relative risks (SRRs) with their corresponding 95% confidence intervals (95% CIs), and assessed the certainty of evidence. In total, 15,070 publications were identified in the literature search after the removal of duplicates. Out of the 180 articles screened in full text, 23 studies (19 cohorts) met our inclusion criteria, with 11 studies (6 cohorts) conducted in the US, 7 studies (7 cohorts) in Europe, 4 studies (5 cohorts) in Asia, and 1 study (1 cohort) in Australia. We mainly observed no or weak linear associations between dietary fats and fatty acids and T2D incidence. In nonlinear dose–response meta-analyses, the protective association for vegetable fat and T2D was steeper at lower levels up to 13 g/d (SRR [95% CI]: 0.81 [0.76; 0.88], $p_{nonlinearity}$ = 0.012, $n$ = 5 studies) than at higher levels. Saturated fatty acids showed an apparent protective association above intakes around 17 g/d with T2D (SRR [95% CI]: 0.95 [0.90; 1.00], $p_{nonlinearity}$ = 0.028, $n$ = 11). There was a nonsignificant association of a decrease in T2D incidence for polyunsaturated fatty acid intakes up to 5 g/d (SRR [95% CI]: 0.96 [0.91; 1.01], $p_{nonlinearity}$ = 0.023, $n$ = 8), and for alpha-linolenic acid consumption up to 560 mg/d

**Data Availability Statement:** All data are available in the manuscript and its Supporting information files.

**Funding:** The German Diabetes Center (DDZ) is funded by the German Federal Ministry of Health and the Ministry of Innovation, Science, Research and Technology of the State North Rhine-Westphalia. The funding source has no role in the decisions about the data collection, analysis, interpretation of the data, preparation, review or approval of the manuscript.

**Competing interests:** The authors have declared that no competing interests exist.

**Abbreviations:** RCT, randomized controlled trial; RR, relative risk; SRR, summary relative risk; T2D, type 2 diabetes.

(SRR [95% CI]: 0.95 [0.90; 1.00], $p_{nonlinearity}$ = 0.014, $n$ = 11), after which the curve rose slightly, remaining close to no association. The association for long-chain omega-3 fatty acids and T2D was approximately linear for intakes up to 270 mg/d (SRR [95% CI]: 1.10 [1.06; 1.15], $p_{nonlinearity}$ < 0.001, $n$ = 16), with a flattening curve thereafter. Certainty of evidence was very low to moderate. Limitations of the study are the high unexplained inconsistency between studies, the measurement of intake of dietary fats and fatty acids via self-report on a food group level, which is likely to lead to measurement errors, and the possible influence of unmeasured confounders on the findings.

## Conclusions

There was no association between total fat intake and the incidence of T2D. However, for specific fats and fatty acids, dose–response curves provided insights for significant associations with T2D. In particular, a high intake of vegetable fat was inversely associated with T2D incidence. Thus, a diet including vegetable fat rather than animal fat might be beneficial regarding T2D prevention.

## Author summary

### Why was this study done?

- Type 2 diabetes is one of the most common noncommunicable diseases worldwide, with a substantial global health burden and healthcare costs.

- The role of dietary fat and fatty acid intake in type 2 diabetes prevention is under debate.

- It was our aim to investigate the associations of dietary intakes of total fat, animal fat, vegetable fat, and various types of fatty acids with type 2 diabetes incidence in an updated systematic review and dose–response meta-analysis of prospective observational studies, and to evaluate the certainty of evidence.

### What did the researchers do and find?

- This systematic review and dose–response meta-analysis included 23 prospective observational studies in adults (≥18 years) on the associations between intake of dietary fats and fatty acids and type 2 diabetes incidence.

- In linear dose–response meta-analyses, no or weak associations between intake of dietary fats and fatty acids and type 2 diabetes incidence were found.

- In nonlinear dose–response meta-analyses, a decrease in type 2 diabetes incidence was associated with higher quantities of vegetable fat in the diet, as well as with lower doses of polyunsaturated fatty acids, including the plant-based fatty acid alpha-linolenic acid. A harmful association of saturated fatty acids with T2D incidence was not confirmed.

### What do these findings mean?

- These findings add to recent evidence that contradicts the long-held belief that diets high in fat increase the risk of type 2 diabetes.

- Additionally, they highlight the importance of the fat source, and specifically reveal an inverse association between plant-based fat intake and type 2 diabetes.

- The findings are limited by very low to moderate certainty of evidence.

## Introduction

Diabetes mellitus is a global health burden with a worldwide prevalence of 9% [1]. Diabetes is characterized by a chronic state of hyperglycemia [2,3]. Type 2 diabetes is the most common type of diabetes (T2D) and accounts for approximately 90% of all cases [1]. In T2D, beta-cell mass and function are lost progressively based on an initial state of insulin resistance [2–4]. T2D increases the risk for diabetes-related complications (e.g., coronary heart disease, stroke, diabetic nephropathy) [5], comorbidities (e.g., depression) [6], and premature death [1,7], and thus leads to higher healthcare costs [1,8]. Apart from unmodifiable risk factors, such as age and family history of diabetes [1,2], several lifestyle-related factors, including smoking, overweight and (abdominal) obesity, and physical activity affect the onset of T2D [9]. Furthermore, diet is a key modifiable factor in the prevention of T2D [10–12]. In this context, the role of dietary fats and fatty acids in T2D prevention is debated [13]. Dietary fats include a wide range of fatty acids, with different chemical structures and biological functions, that play an important role in metabolic pathways influencing the risk of T2D [14]. Current dietary guidelines on the prevention of T2D recommend a diet low in total fat and animal fat, and high in vegetable fat [11,12]. Additionally, higher intakes of monounsaturated fatty acids [12,15], polyunsaturated fatty acids [12,15], and omega-3 fatty acids [12], as well as lower intakes of saturated fatty acids [11] and *trans*-fatty acids [12], are recommended. While results of meta-analyses have indicated a protective association of vegetable fat intake with T2D incidence, the intake of single types of fatty acids, such as saturated fatty acids, monounsaturated fatty acids, and polyunsaturated fatty acids, was not associated with incidence of T2D [16]. However, these meta-analyses summarized prospective cohort studies published up to the year 2014 [17–20], and new prospective cohort studies examining the associations between dietary fat and fatty acid intake have recently been published [21–27]. Moreover, dose–response relationships have not yet been examined for the majority of these associations. Thus, an updated systematic review and dose–response meta-analysis are necessary. Additionally, a certainty of evidence assessment for these updated meta-analyses is warranted.

Therefore, our first aim was to examine the associations between dietary intakes of total fat, animal fat, vegetable fat, and various types of fatty acids (saturated fatty acids, monounsaturated fatty acids, polyunsaturated fatty acids [including omega-6 and omega-3 fatty acids], and *trans*-fatty acids) and T2D incidence in an updated systematic review and dose–response meta-analysis of prospective observational studies in an adult population. Second, we aimed to evaluate the certainty of evidence for these associations.

## Methods

Our protocol was prospectively registered at PROSPERO (CRD42019128664). We followed the Preferred Reporting Items for Systematic Reviews and Meta-Analyses (PRISMA) guidelines [28] (see S1 PRISMA Checklist).

## Study search and selection

PubMed, Web of Science, and reference lists of relevant publications were systematically searched from their starting dates to 28 October 2019 applying no restrictions or filters. The following search terms were used in combination: (fat OR fats OR fatty OR "fish oil" OR "fish oils") AND diabetes AND ("observational study" OR prospective OR cohort OR cohorts OR longitudinal OR "case-control" OR retrospective OR "follow-up").

The literature search and study selection were conducted by 3 investigators independently (MN, NI, and JBe). Disagreements were solved via discussion until consensus was reached. Studies were included if they met the following criteria: (1) prospective observational studies (cohort studies, nested case–control studies, case–cohort studies, follow-up of randomized controlled trials [RCTs]), (2) main focus on adults (≥18 years), (3) reported on associations between intake of total fat, animal fat, vegetable fat, or types of fatty acids (e.g., saturated fatty acids, monounsaturated fatty acids, polyunsaturated fatty acids) and incidence of T2D, and (4) provided effect estimates, reported as hazard ratios, relative risks (RRs), or odds ratios, with corresponding 95% confidence intervals (CIs).

Studies including children, adolescents, pregnant women, individuals with diabetes at baseline, or specific patient groups (e.g., patients after myocardial infarction), as well as animal studies and studies investigating fatty acids measured as biomarkers in plasma/serum, were excluded.

## Data extraction

Data extraction was conducted by one author (MN) and double-checked by a second author (JBa). The following characteristics were extracted from each study: last name of the first author, year of publication, the country where the study was conducted, the cohort name (if any), duration of follow-up, characteristics of the cohort at baseline (age, sex), total number of participants, number of cases of T2D, outcome assessment (self-report of diabetes with or without objective medical details, use of diabetes medication, blood test, medical records), exposure (total fat, animal fat, vegetable fat, types of fatty acids), exposure assessment (questionnaire with or without validation, interviews), fat or fatty acid intake per category, person-years and number of cases per category, and maximally adjusted risk estimates expressed as hazard ratios, RRs, or odds ratios with corresponding 95% CIs and adjustment factors. If important data were missing, we contacted the authors of the original studies for more information.

## Risk of bias assessment

Risk of bias assessment for each study was conducted by 2 investigators (MN and LS) independently, using the Cochrane Risk of bias in Non-randomized Studies of Interventions (ROBINS-I) tool [29]. The tool includes 7 domains of bias due to (1) confounding, (2) selection of participants, (3) exposure assessment, (4) misclassification of exposure during follow-up, (5) missing data, (6) measurement of the outcome, and (7) selective reporting of the results. The detailed description of each potential risk of bias domain is provided in S1 Table. Discrepancies were resolved by consensus or the consultation of a third reviewer (SS).

## Certainty of evidence assessment

Additionally, we evaluated the certainty of evidence for each association using the updated Grading of Recommendations Assessment, Development and Evaluations (GRADE) [30] system, which integrates the application of ROBINS-I. In contrast to the previous version [31],

observational studies also start at a high certainty of evidence level [30]. However, a lack of randomization leads to a downgrading by 2 levels (to low), unless the study design reduces confounding and selection bias, as evaluated by ROBINS-I. Additionally, indications of inconsistency, indirectness, imprecision, and publication bias can lead to downgrading, while large effects and a dose–response gradient can lead to upgrading [30,31]. High and moderate certainty of evidence mean that it is very likely or probable that the true effect lies close to the estimated effect. Our confidence in the result is limited or weak if the certainty of evidence is rated as low or very low, respectively [31].

## Statistical analysis

We calculated summary RRs (SRRs) using a random effects model, taking both within- and between-study variability into account [32]. The average of the natural logarithm of the RRs was estimated, and the RR from each study was weighted using the method of moments by DerSimonian and Laird [33]. We conducted linear dose–response meta-analyses using the method by Greenland and Longnecker [34]. We computed study-specific slopes (linear trends) and 95% CIs based on the natural logarithm of the RRs and 95% CIs across categories of each exposure (total fat, animal fat, vegetable fat, various types of fatty acids). For this analysis, the number of cases and person-years per category and the exposure values with RRs and corresponding 95% CIs of at least 3 categories were needed. If not reported, the distribution of cases and person-years was estimated, using information on the total number of cases and the number of total participants plus the follow-up period as previously described elsewhere [35]. If a study reported the exposure categories as ranges, the midpoint between the lower and the upper limit was calculated for each category. For open categories, a similar range to the adjacent category was assumed. If the dietary fat or fatty acid dose per category was not reported in grams per day but as percent of total energy intake, we converted energy percent into grams per day. We calculated the calories of the dietary fat/fatty acid by multiplying energy percent by the mean energy intake in the cohort. In order to estimate grams per day, we divided the calories of this dietary fat/fatty acid by 9.1 kcal, which is the amount of calories provided by 1 gram of fat intake. If mean total energy intake of the cohort was not reported in the publication [36,37], information from another publication of the same cohort was used [38–41]. The doses for the linear dose–response meta-analyses were chosen as previously described [18,42]. Nonlinear dose–response meta-analyses were conducted using a restricted cubic spline model as described by Orsini et al. [43], with 3 knots at the 10th, 50th, and 90th percentile of frequency of each exposure. We used a likelihood ratio to test for nonlinearity, checked goodness of fit ($\chi^2$) for the nonlinear model compared to the linear model, and interpreted the curve based on visual inspection of the graph.

To assess potential heterogeneity, we conducted subgroup analyses stratified by sex, geographic location, duration of follow-up, number of cases, exposure assessment, outcome assessment, quality score, and adjustment for confounding factors, and applied meta-regression analysis [44]. Furthermore, we conducted sensitivity analyses omitting 1 study at a time to investigate the influence of each study on the results.

We calculated $I^2$ and $\tau^2$ as measures of the inconsistency and between-study variability of the risk estimates, respectively, and computed 95% prediction intervals (95% PIs), which show the range in which the underlying true effect of future studies will lie with 95% certainty [45,46].

Publication bias and small study effects were assessed using funnel plots and Egger's test [47,48] if at least 10 studies were available, as recommended by Cochrane [49]. Potential publication bias was indicated by asymmetry of the funnel plot and a $p$-value of <0.1 for Egger's test [48].

All statistical analyses were conducted using STATA version 14.1.

## Results

In total, 23 studies (19 cohorts) met our inclusion criteria (S1 Fig). Excluded studies with respective exclusion reasons are displayed in S2 Table. The characteristics of the included studies are summarized in S3 Table. Eleven studies (6 cohorts) were conducted in the US [25–27,37,50–56], 7 studies (7 cohorts) in Europe [21–23,36,57–59], 4 studies (5 cohorts) in Asia [24,60–62], and 1 study (1 cohort) in Australia [63]. All studies used validated food frequency questionnaires for the exposure assessment, except for 2 studies that used 3- or 4-day food records [57,59]. Four studies validated the dietary intakes of fatty acids against biomarkers measured in adipose tissue [37,52,64] and erythrocyte membranes [36] and reported weak to moderate correlations (Spearman correlation coefficients between ≤0.19 and 0.51) [36,37,52,64]. All studies adjusted for age, sex, smoking, education, and total energy intake, except for 2 studies that did not adjust for education [51] or for education, smoking, and total energy intake [57].

Twenty studies were judged as being at moderate risk of bias, and 3 studies as being at serious risk of bias, due to insufficient adjustment of relevant confounders, as described above (S4 Table). Generally, risk of bias due to confounding and exposure assessment could never be low, because of the possibility of residual confounding in observational studies and measurement error in the dietary assessment.

Fig 1 summarizes the results of the linear dose–response meta-analyses for each type of fat and fatty acid regarding T2D incidence. Forest plots of all meta-analyses for each exposure are displayed in S2 and S3 Figs. In these analyses, we mainly observed no or weak associations between dietary fat and fatty acid intake and T2D incidence.

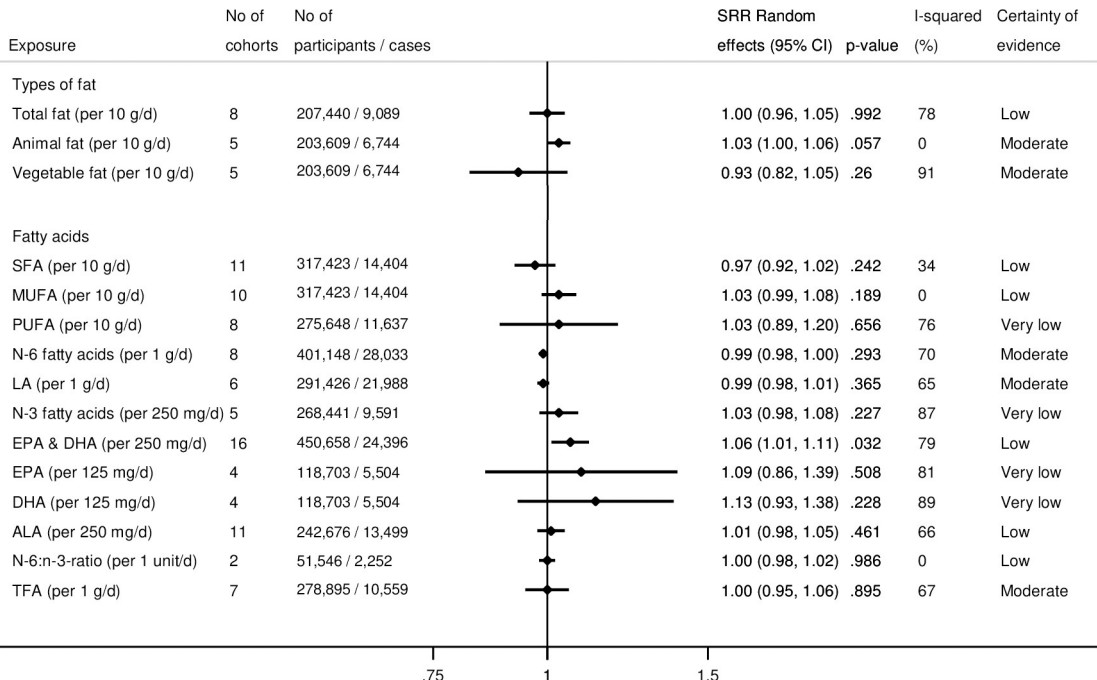

| Exposure | No of cohorts | No of participants / cases | SRR Random effects (95% CI) | p-value | I-squared (%) | Certainty of evidence |
|---|---|---|---|---|---|---|
| **Types of fat** | | | | | | |
| Total fat (per 10 g/d) | 8 | 207,440 / 9,089 | 1.00 (0.96, 1.05) | .992 | 78 | Low |
| Animal fat (per 10 g/d) | 5 | 203,609 / 6,744 | 1.03 (1.00, 1.06) | .057 | 0 | Moderate |
| Vegetable fat (per 10 g/d) | 5 | 203,609 / 6,744 | 0.93 (0.82, 1.05) | .26 | 91 | Moderate |
| **Fatty acids** | | | | | | |
| SFA (per 10 g/d) | 11 | 317,423 / 14,404 | 0.97 (0.92, 1.02) | .242 | 34 | Low |
| MUFA (per 10 g/d) | 10 | 317,423 / 14,404 | 1.03 (0.99, 1.08) | .189 | 0 | Low |
| PUFA (per 10 g/d) | 8 | 275,648 / 11,637 | 1.03 (0.89, 1.20) | .656 | 76 | Very low |
| N-6 fatty acids (per 1 g/d) | 8 | 401,148 / 28,033 | 0.99 (0.98, 1.00) | .293 | 70 | Moderate |
| LA (per 1 g/d) | 6 | 291,426 / 21,988 | 0.99 (0.98, 1.01) | .365 | 65 | Moderate |
| N-3 fatty acids (per 250 mg/d) | 5 | 268,441 / 9,591 | 1.03 (0.98, 1.08) | .227 | 87 | Very low |
| EPA & DHA (per 250 mg/d) | 16 | 450,658 / 24,396 | 1.06 (1.01, 1.11) | .032 | 79 | Low |
| EPA (per 125 mg/d) | 4 | 118,703 / 5,504 | 1.09 (0.86, 1.39) | .508 | 81 | Very low |
| DHA (per 125 mg/d) | 4 | 118,703 / 5,504 | 1.13 (0.93, 1.38) | .228 | 89 | Very low |
| ALA (per 250 mg/d) | 11 | 242,676 / 13,499 | 1.01 (0.98, 1.05) | .461 | 66 | Low |
| N-6:n-3-ratio (per 1 unit/d) | 2 | 51,546 / 2,252 | 1.00 (0.98, 1.02) | .986 | 0 | Low |
| TFA (per 1 g/d) | 7 | 278,895 / 10,559 | 1.00 (0.95, 1.06) | .895 | 67 | Moderate |

SFA: saturated fatty acids; MUFA: monounsaturated fatty acids; PUFA: polyunsaturated fatty acids; n-6 fatty acids: omega-6 fatty acids; LA: linoleic acid; n-3 fatty acids: omega-3 fatty acids; EPA: eicosapentaenoic acid; DHA: docosahexaenoic acid; ALA: alpha-linolenic acid; TFA: *trans*-fatty acids

**Fig 1. Summary relative risks (SRRs) with 95% confidence intervals (95% CIs) for the associations of total fat, animal fat, vegetable fat, and different fatty acids with incidence of type 2 diabetes in linear dose–response meta-analyses.**

However, we detected nonlinear associations for specific fats and fatty acids (Figs 2–5). We observed a steep significant association with a decrease in T2D incidence up to a 13-g/d intake of vegetable fat (SRR [95% CI]: 0.81 [0.76; 0.88], $p_{nonlinearity}$ = 0.012; goodness of fit: $\chi^2_{nonlinear}$ = 47.4 versus $\chi^2_{linear}$ = 37.1), after which the curve almost reached a plateau (Fig 2C). Regarding saturated fatty acids, the curve declined after a dose of 8 g/d (SRR [95% CI]: 1.02 [0.97; 1.07]), with an apparent association with a decrease in T2D incidence for intakes around 17 g/d (SRR [95% CI]: 0.95 [0.90; 1.00], $p_{nonlinearity}$ = 0.028; goodness of fit: $\chi^2_{nonlinear}$ = 39.7 versus $\chi^2_{linear}$ = 15.2) (Fig 3A). For polyunsaturated fatty acids, doses up to 5 g/d were nonsignificantly associated with reduced T2D incidence (SRR [95% CI]: 0.96 [0.91; 1.01], $p_{nonlinearity}$ = 0.023; goodness of fit: $\chi^2_{nonlinear}$ = 42.1 versus $\chi^2_{linear}$ = 29.6), after which the curve rose slightly, remaining close to no association (Fig 3C). We observed a steep significant association with a rise in T2D incidence up to an intake of 270 mg of long-chain omega-3 fatty acids (SRR [95% CI]: 1.10 [1.06; 1.15], $p_{nonlinearity}$ = <0.001; goodness of fit: $\chi^2_{nonlinear}$ = 105.7 versus $\chi^2_{linear}$ = 70.9), with a more modest association with increased T2D incidence thereafter (Fig 5C). The curves for eicosapentaenoic acid and docosahexaenoic acid showed an inverse U-shape, with a steep, but nonsignificant, association with a rise in T2D incidence up to intakes of 110 mg/d and 200 mg/d, respectively (Fig 5D and 5E). Regarding alpha-linoleic acid, we observed a flat J-shaped relation, with an apparent association with a decrease in T2D incidence up to an alpha-linolenic acid intake of 560 mg/d (SRR [95% CI]: 0.95 [0.90; 1.00], $p_{nonlinearity}$ = 0.014; goodness of fit: $\chi^2_{nonlinear}$ = 54.6 versus $\chi^2_{linear}$ = 29.0), after which the curve moderately rose, remaining close to no association (Fig 5B).

## Certainty of evidence

No association was rated as having a high certainty of evidence. We found moderate, low, and very low certainty of evidence for 5, 6, and 4 associations, respectively (Figs 1 and S5). This judgment was mainly driven by concerns regarding risk of bias due to the possibility of residual confounding, inconsistency, and indirectness.

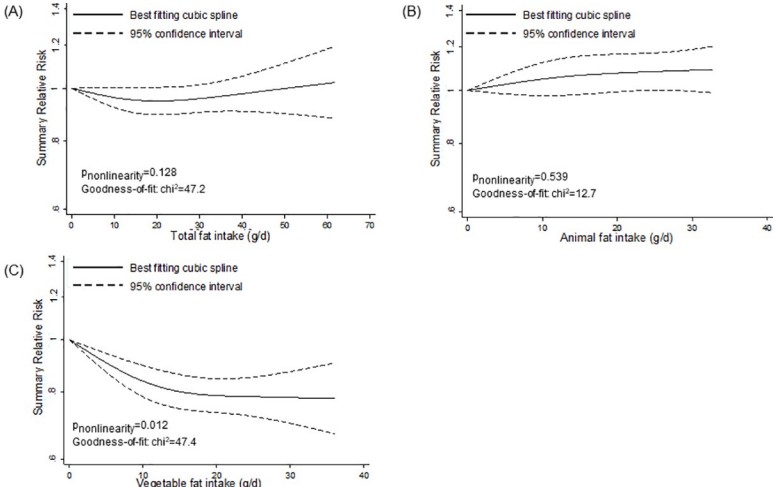

**Fig 2. Nonlinear dose–response meta-analyses for the associations between dietary fats and incidence of type 2 diabetes.** (A) Total fat. (B) Animal fat. (C) Vegetable fat.

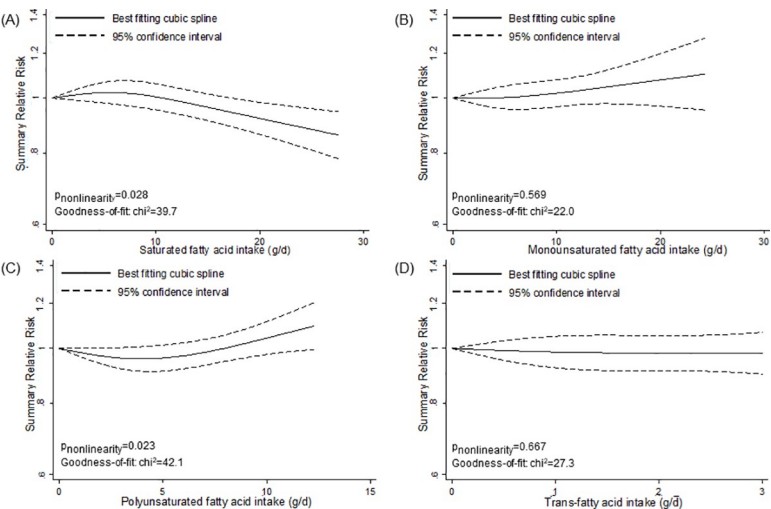

**Fig 3. Nonlinear dose–response meta-analyses for the associations between types of fatty acids and incidence of type 2 diabetes.** (A) Saturated fatty acids. (B) Monounsaturated fatty acids. (C) Polyunsaturated fatty acids. (D) *Trans-*fatty acids.

## Subgroup and sensitivity analysis

S6 Table and S4 and S5 Figs display the results of the subgroup and sensitivity analyses, respectively. Most of the results were robust in both analyses. However, important geographical differences were observed regarding long-chain omega-3 fatty acids. The association was attenuated in European studies but was stronger in US populations. Contrary to the main analysis, an inverse association between long-chain omega-3 fatty acids and T2D incidence was observed in Asian populations (S6 Table). These differences were also apparent in nonlinear dose–response meta-analyses stratified by geographic location (S6 Fig).

In sensitivity analyses, based on the stepwise omission of 1 study at a time, the exclusion of the PREDIMED study [23] led to a reduced and more precise estimate for vegetable fat (S4

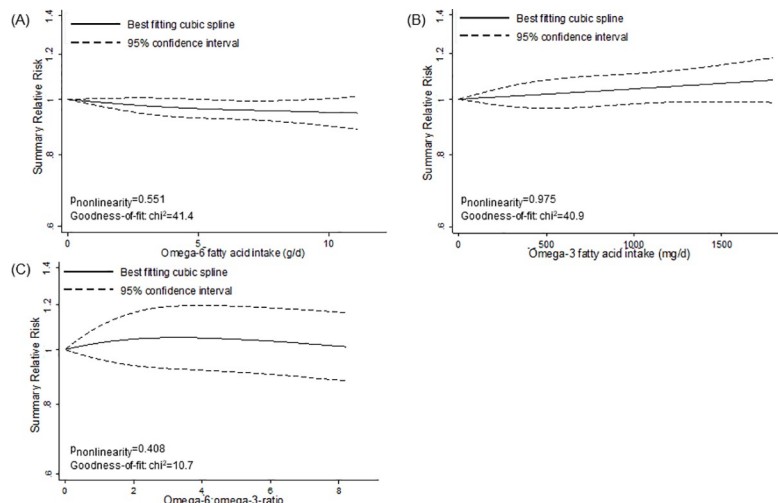

**Fig 4. Nonlinear dose–response meta-analyses for the associations between omega-6 and omega-3 fatty acids and incidence of type 2 diabetes.** (A) Omega-6 fatty acids. (B) Omega-3 fatty acids. (C) Omega-6:omega-3 ratio.

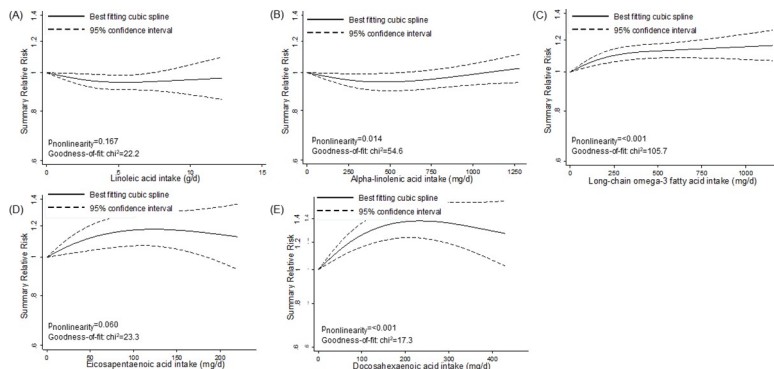

**Fig 5. Nonlinear dose–response meta-analyses for the associations between specific fatty acids and incidence of type 2 diabetes.** (A) Linoleic acid. (B) Alpha-linolenic acid. (C) Long-chain omega-3 fatty acids. (D) Eicosapentaenoic acid. (E) Docosahexaenoic acid.

Fig), while the exclusion of the Nurses' Health Study [37] led to an association with reduced T2D incidence for *trans*-fatty acids (S5 Fig).

## Small study effects and publication bias

Ten or more studies were available for saturated fatty acids, monounsaturated fatty acids, long-chain omega-3 fatty acids, and alpha-linolenic acid. There was no indication for small study effects according to the funnel plots or the Egger's test for these associations (S7 Fig). However, for long-chain omega-3 fatty acids and alpha-linolenic acid, the funnel plots indicated between-study variability due to values outside of the 95% confidence limits.

## Discussion

In this systematic review and dose–response meta-analysis, we observed an association with decreased T2D incidence for higher intake of vegetable fat, especially of plant-based alpha-linolenic acid, and for total polyunsaturated fatty acids in lower doses. Animal-based long-chain omega-3 fatty acids were associated with increased T2D incidence; however, geographic differences were observed. A harmful association for saturated fatty acids was not confirmed. Our findings add to the body of evidence that these associations are not linear. Most studies were of moderate risk of bias, and the certainty of evidence was very low to moderate.

Our results are consistent with findings from previous meta-analyses investigating the associations of high versus low fat and fatty acid intakes with T2D incidence [17,19,20,65,66] and do not support guidelines recommending increased intake of monounsaturated fatty acids [15], total omega-3 fatty acids [12], or long-chain omega-3 fatty acids [15], or lower intake of saturated fatty acids [11] and *trans*-fatty acids [12], for T2D prevention. In line with previous results of high versus low intake meta-analysis [17], our nonlinear dose–response meta-analysis indicated a significant association of decreased T2D incidence with increasing vegetable fat intake. A recent meta-analysis of RCTs [65] found increased T2D incidence with higher omega-6 fatty acid intake and an inverse association for alpha-linolenic acid intake [65]. However, these associations were very imprecisely estimated, based on only 2 trials, and the control groups of the trials varied, including comparisons with mixed fat intake or low doses of the same fatty acid [65]. We observed geographic differences regarding long-chain omega-3 fatty acids, which were also found by Wallin and colleagues [67] and in biomarker studies [62,68]. Previous high versus low intake meta-analyses observed an inverse, though nonsignificant, association between alpha-linolenic acid intake and T2D incidence [17,18]. In our nonlinear

dose–response meta-analysis, an apparent association with decreased T2D incidence was observed for an intake of up to 560 mg alpha-linolenic acid per day. However, this estimate was also imprecise. Regarding *trans*-fatty acids, our null findings confirm the results of 1 high versus low intake meta-analysis [17], but not another [20]. Differences might be explained by different compositions of *trans*-fatty acids because neither meta-analysis differentiated between industrial and ruminant *trans*-fatty acids. However, biomarker studies showed that ruminant *trans*-fatty acids were associated with decreased T2D incidence [20], while industrial *trans*-fatty acids increased T2D incidence [69]. However, our results are supported by findings of previously conducted RCTs suggesting no effect on glucose metabolism when comparing diets high in total *trans*-fatty acids to diets low in total *trans*-fatty acids [70].

In general, individuals following an unhealthy diet (e.g., high animal fat intake through meat consumption) are also likely to have an unhealthy lifestyle (e.g., higher rates of obesity and smoking and lower levels of physical activity) [71,72]. Despite the adjustment for body mass index (BMI), smoking, and physical activity, residual confounding is possible. However, recent evidence does not support the long-held belief that high fat diets lead to obesity, and thus, T2D. In contrast, apart from being an energy source, fatty acids also have important bio-active properties [14]. Moreover, it is likely that any association of fat or fatty acid intake with T2D incidence depends on the overall dietary pattern and the food source [14,73]. For example, olive oil is associated with decreased T2D incidence [74], while the health effects of other vegetable fats, such as palm oil and coconut oil, are debated [14]. Vegetable fats also include plant-derived polyunsaturated fatty acids. In a meta-analysis of RCTs, higher intakes of plant-based polyunsaturated fatty acids showed beneficial effects on insulin resistance (HOMA-IR) and fasting insulin levels compared to higher intakes of carbohydrate or saturated fatty acids [75]. Additionally, there is indication that lower levels of plant-derived alpha-linolenic acid are associated with higher pro-inflammatory markers [76] and therefore influence inflammatory processes playing an important role in the development of T2D [77,78]. In contrast, depending on geographic location, higher intakes of animal-based long-chain omega-3 fatty acids were associated with increased T2D incidence. In this context, differences in food preparation between countries might play a role [14]. In addition, investigations into the influence of genetic susceptibility on the association between fat and fatty acid intake and T2D incidence have not yielded consistent insight [14,79–81]. Therefore, further research examining the role of genetic susceptibility is warranted [14]. The different food sources and structures of fatty acids play an important role as well. Saturated fatty acids are contained in meat products, including red and processed meat, which are associated with increased T2D incidence [16]. Additionally, short-chain, even-chain saturated fatty acids increase T2D incidence [82,83]. However, saturated fatty acids are also contained in dairy products, as well as in low concentrations in peanuts and canola oil, which are sources of odd-chain and very-long-chain saturated fatty acids, respectively, which decreased T2D incidence in biomarker studies [82–84]. Dairy products are also a source of ruminant *trans*-fatty acids, which are produced by bacterial metabolism of polyunsaturated fatty acids in the stomach of ruminants [85] and which were associated with decreased T2D incidence [20]. In contrast, industrial *trans*-fatty acids from processed food products have been shown to be associated with increased T2D incidence [69]. And although all studies included in our meta-analyses adjusted for additional dietary factors, such as other fatty acids, none of these studies investigated the food sources, for example animal versus plant products, in their analyses. Additionally, the nutrient composition of the whole diet plays a role. For example, substitution studies indicate that an isocaloric replacement of carbohydrates with saturated fatty acids is associated with decreased T2D incidence [79]. However, replacing saturated fatty acids or carbohydrates with polyunsaturated fatty acids lowered fasting glucose levels and glycated hemoglobin (HbA1C) and improved insulin

resistance (HOMA-IR), but did not affect fasting glucose or postprandial glucose and insulin levels [86].

## Strengths and limitations

To our knowledge, this is the first dose–response meta-analysis that provides a comprehensive overview of all associations between dietary fat and fatty acid intake and T2D incidence, including extensive subgroup and sensitivity analyses. Additionally, we assessed the risk of bias of each included study and evaluated the certainty of evidence for each association using validated tools. Because we only included prospective studies, risk of recall and selection bias was reduced.

However, our study also has a number of limitations. For half of the exposures, only 5 or fewer studies were available for the meta-analyses. Therefore, subgroup analyses of these associations were only based on a few studies or were not possible at all. Moreover, publication bias could only be assessed for saturated fatty acids, monounsaturated fatty acids, long-chain omega-3 fatty acids, and alpha-linolenic acid. Additionally, most of the observed high inconsistency between the studies remained unexplained, leading to lower certainty of evidence. This might be due to different fatty acid compositions of the fatty acid classes (e.g., differences between the studies regarding the proportions of even-chain and odd-chain saturated fatty acids in total saturated fatty acids). The applied conventional classification into groups of fat (e.g., vegetable fat) and classes of fatty acids (e.g., saturated fatty acids) might conceal differences regarding bioactive properties of different fatty acids within each group and class [14]. Investigating finer strata of these classes in biomarker studies might provide further insights. Additionally, since dietary fat intake was assessed via self-reports, measurement errors are likely. Moreover, in food frequency questionnaires, only the main food sources for fatty acids are included, and they are assessed on a food group level, which might lead to difficulties in quantifying fat and fatty acid intake. Only 4 of the included studies validated the dietary intakes of fatty acids measured via food frequency questionnaires against biomarkers, and these studies reported weak to moderate correlations. Biomarker studies might therefore add a more objective and reliable measure, especially for omega-6 and omega-3 fatty acids [14]. Furthermore, most studies provided no information on the main food sources contributing to fat and fatty acid intake. However, the food sources play a major role, especially for the interpretation of the results regarding saturated fatty acids, monounsaturated fatty acids, and possibly *trans*-fatty acids [14]. Such uncertainties contributed to the downgrading regarding the certainty of evidence. Therefore, future studies should also investigate the role of different food sources in relation to the association of fats and fatty acids with T2D incidence. Moreover, we observed geographic differences in the association of T2D incidence with long-chain omega-3 fatty acids. Reasons for these differences are not yet clear, and more research regarding the possible mediating role of genetic susceptibility is warranted. Lastly, since we included observational studies, residual confounding cannot be ruled out.

## Conclusions

In our linear dose–response meta-analyses, we mainly observed no or weak associations between intake of dietary fats and fatty acids and T2D incidence. However, in nonlinear dose–response meta-analyses, we observed a significant association of decreased T2D incidence with higher intakes of vegetable fat, as well as a non-significant decrease in T2D incidence for polyunsaturated fatty acids and alpha-linolenic acid in lower doses. Long-chain omega-3 fatty acids were associated with a significant decrease in incidence of T2D in Asian populations, and with a significant increase in incidence of T2D in US populations. A harmful association

for saturated fatty acids was not confirmed. However, our results are limited by very low to moderate certainty of evidence. To strengthen the evidence, future studies should focus on the association between the fatty acid composition of the diet and T2D. In addition, further research is needed to investigate the role of different food sources regarding the association between fatty acid intake and T2D incidence.

## Supporting information

**S1 PRISMA Checklist. PRISMA checklist.**
(DOC)

**S1 Fig. Flow chart of literature search.**
(DOCX)

**S2 Fig. Linear dose–response meta-analyses of the associations between total fat, animal fat, and vegetable fat and incidence of type 2 diabetes.**
(DOCX)

**S3 Fig. Linear dose–response meta-analyses of the associations between specific fatty acids and incidence of type 2 diabetes.**
(DOCX)

**S4 Fig. Sensitivity analyses for total fat, animal fat, and vegetable fat.**
(DOCX)

**S5 Fig. Sensitivity analyses for specific fatty acids.**
(DOCX)

**S6 Fig.** Nonlinear dose–response meta-analyses for the association between long-chain omega-3 fatty acids and incidence of type 2 diabetes stratified by geographic region.
(DOCX)

**S7 Fig. Funnel plots.**
(DOCX)

**S1 Table. Description and decision criteria for each domain in ROBINS-I.**
(DOCX)

**S2 Table. List of excluded studies.**
(DOCX)

**S3 Table. Study characteristics of the included studies.**
(DOCX)

**S4 Table. ROBINS-I judgment for each domain and overall.**
(DOCX)

**S5 Table. GRADE judgment for each domain and overall.**
(DOCX)

**S6 Table. Linear dose–response meta-analyses by subgroups.**
(DOCX)

## Author Contributions

**Conceptualization:** Manuela Neuenschwander, Sabrina Schlesinger.

**Data curation:** Manuela Neuenschwander, Janett Barbaresko, Nadine Iser, Julia Beckhaus, Lukas Schwingshackl, Sabrina Schlesinger.

**Formal analysis:** Manuela Neuenschwander, Lukas Schwingshackl, Sabrina Schlesinger.

**Investigation:** Manuela Neuenschwander, Janett Barbaresko, Claudia R. Pischke, Nadine Iser, Julia Beckhaus, Lukas Schwingshackl, Sabrina Schlesinger.

**Methodology:** Manuela Neuenschwander, Lukas Schwingshackl, Sabrina Schlesinger.

**Supervision:** Sabrina Schlesinger.

**Writing – original draft:** Manuela Neuenschwander, Sabrina Schlesinger.

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
