## [Editor Report · Decision Letter 0]

6 Apr 2020

Dear Dr Schlesinger, 

Thank you for submitting your manuscript entitled "Intake of dietary fats, fatty acids and the incidence of type 2 diabetes: a systematic review and dose-response meta-analysis of prospective observational studies" for consideration by PLOS Medicine.

Your manuscript has now been evaluated by the PLOS Medicine editorial staff [as well as by an academic editor with relevant expertise] and I am writing to let you know that we would like to send your submission out for external peer review.

Kind regards,

Clare Stone, PhD,

PLOS Medicine

---

## [Decision Letter · Decision Letter 1]

7 May 2020

Dear Dr. Schlesinger,

Thank you very much for submitting your manuscript "Intake of dietary fats, fatty acids and the incidence of type 2 diabetes: a systematic review and dose-response meta-analysis of prospective observational studies" (PMEDICINE-D-20-00813R1) for consideration at PLOS Medicine. 

[LINK]

In light of these reviews, I am afraid that we will not be able to accept the manuscript for publication in the journal in its current form, but we would like to consider a revised version that addresses the reviewers' and editors' comments. Obviously we cannot make any decision about publication until we have seen the revised manuscript and your response, and we plan to seek re-review by one or more of the reviewers. 

We expect to receive your revised manuscript by May 28 2020 11:59PM. Please email us (plosmedicine@plos.org) if you have any questions or concerns.

We look forward to receiving your revised manuscript. 

Sincerely,

Emma Veitch, PhD

PLOS Medicine

On behalf of Clare Stone, PhD, Acting Chief Editor,

PLOS Medicine

plosmedicine.org

*Please structure your abstract using the PLOS Medicine headings (Background, Methods and Findings, Conclusions; Methods and Findings sections combined into one single subsection). Please also make sure the abstract subsections are written in complete sentences, not sentence fragments/bullet type points. The methodological detail included currently in the abstract is at an appropriate level though so would recommend this level of detail is kept.

*At this stage, we ask that you include a short, non-technical Author Summary of your research to make findings accessible to a wide audience that includes both scientists and non-scientists. The Author Summary should immediately follow the Abstract in your revised manuscript. This text is subject to editorial change and should be distinct from the scientific abstract. Please see our author guidelines for more information: https://journals.plos.org/plosmedicine/s/revising-your-manuscript#loc-author-summary

*If possible, please update the in-text callouts to references to numbered callouts in square brackets (eg [1, 2]), if you've used referencing software this should be fairly simple - many thanks.

*Currently the study reporting has used ROBINS-I and GRADE to evaluate study quality, which is good, but it doesn't seem that a reporting tool (such as PRISMA) was used for the entirety of the systematic review (in general). This point was raised by reviewers, it would be good to address this by using the PRISMA tool to update any details of methodological/findings reporting within the manuscript, and then provide the completed PRISMA checklist as a supporting information file. When completing the checklist, please use section and paragraph numbers, rather than page numbers. 

*Minor typos/grammatical points - eg, Page 19 (Discussion), "recent evidence does not support the long-time believe that high fat diets"  long-term belief

*One reviewer has asked authors to change use of the first person ("we") in the paper - the editors would disagree, and although this can be considered a bit colloquial in scientific manuscripts, we prefer this type of direct writing. 

Comments from the reviewers:

Reviewer #1: I confine my remarks to statistical aspects of this paper.

Mostly, everything was find but I have a couple issues to resolve before I can recommend publication.

Line 120-123 were unclear to me. What exactly was done?

While I like the use of nonlinear models, I don't think a p value (line 125) is the right way to judge. Instead, look at the differences in the models and make a judgement as to whether the complexity of splines is made up for by improvements in fit.

For the forest plot, I'd like to see more of the area devoted to the graph. In the first column, one line is much longer than all the others, it could be wrapped. Then the second column (dose) could be merged with the first. I[m not sure all of the next three columns are needed. Some might be put in footnotes.

Peter Flom

Reviewer #2: General: 

This is a well conducted systematic review of associations between fat quality and incidence of T2D. Clearly written and presented. Methods appear robust. Bias and limitations of these observational studies is made clear. Certainly of evidence is determined and presented.

Specific:

Abstract - please include details of total number of articles retrieved using key words and total number included in the review.

Introduction. Line 70. I do not understand the statement used to justify conduct of this 2nd systematic review, following rapidly on the back of the author's previous 2019 BMJ umbrella review: 'Because new prospective cohort studies examining the associations between dietary fat and fatty acid intake have recently been published (10-16)'. The BMJ analysis was conducted using trials published up to August 2018. Yet the additional references quoted (refs 10-16) as updating that review were published between 2015 - 2019 (Dow et al. BJN. 2016;Ericson et al. AJCN. 2015; Guasch-Ferre et al. AJCN. 2017:Ha et al. Diab Res Clin Pract. 2019; Ma et al. AJCN. 2015 etc). This is not correct, and requires clarification. Clearly this review is differentiated and warranted, but the justification provided appears confused.

I suggest that some trial methodology information is provided in the Abstract. The findings of this review are arguably unexpected. It is important that readers have ready access to limitations (or strengths) of the 1ry cohort data. Eg. Line 295-7: dietary fat was assessed via self-report, FFQs were the main assessment tool (90% of studies reported), food group level used for assessment.

Line 72-3. Also, it is stated by the current authors that certainty of evidence …. has not been evaluated'; yes contrary to this their earlier BMJ analysis states that a quality evaluation was conducted 'The quality of evidence was evaluated by using a modified version of NutriGrade'. This also requires clarification.

Methods. The methods are valid, and the analysis looks to be well conducted. The protocol was prospectively registered on PROSPERO, as required. Risk of bias was also correctly handled using Cochrane tools. However, some questions still arise

Was the data extracted and checked by 2 independent reviewers?

Results. Line 143. States that 23 studies met the inclusion criteria. However only 22 studies (11+6+4+1) are shown as included, lines 145-7. Why is there a discrepancy?

Also, data on number of European cohorts is missing. Total cohorts = 19? Only 11 are reported in the expanded text.

Statistical analyses. I would recommend including brief description of methods from references #24,25,26, for the linear dose-response metanalyses.

Figures 2 and 3. I would recommend higher quality figures for publication. Also ensure that the y axis is consistent for all figures within a single multi-panel plot, eg. Fig 3, PUFA plot.

Reviewer #3: The manuscript entitled: "Intake of dietary fats, fatty acids and the incidence of type 2 diabetes: a systematic review and dose-response meta-analysis of prospective observational studies" by Neuenschwander et al., describes the impact of various fats and fatty acids in intake and the relationship with diabetes incidence which is an important issue. Yet, the current manuscript confirms results from previous meta-analyses. However, if authors added some supplementary analysis from subgroups or secondary outcomes (such as diabetes risk factors), the manuscript could be more novel. In addition, the English style/grammar of the manuscripts needs to be adjusted. Please see specific comments below.

General: 

There are a few typos and grammar mistakes which should be revised. Please avoid using "we" in the manuscript.

Please add the definition for each abbreviation that you write for the first time and use abbreviations after first in the text. For example, type 2 diabetes (T2D). 

Title: 

 Intake of dietary fats, fatty acids and the incidence of type 2 diabetes should be modified to Intake of dietary fats and fatty acids with the incidence of type 2 diabetes. Because fatty acids could also mean in serum/plasma, it is better to specify that it is intake.

Abstract

Line 33: associations between dietary fat, fatty acids and T2D= please rewrite to association between dietary fat and fatty acid intake with T2D.

Conclusion: Especially…. Please re-write the sentence not starting by especially, perhaps "Specifically" 

Introduction:

Please define diabetes in the first paragraph and then type 2 diabetes briefly. Perhaps adding the risk factors. 

Line 59: What other problems, please be specific.

Line 67-68: Please give general fat (total, vegetable and animal fat) recommendations for individuals with T2D. The study is about dietary fats more information about any single fat in introduction gives more insight to the readers. 

Lines 67-68: Based on recent studies all trans fatty acids may not be harmful. Please explain briefly about industrial and ruminant trans fatty acids. Did the researchers consider the different types of trans-fats in their analyses? 

Line 68: What fatty acids? Do you mean trans fats and saturated fats? Please be specific.

Line 74-79: Please write in 2 sentences. 

Method:

Line 89: It would be better if you add a PRISMA for your paper, it makes easier for the readers to understand study search and selection.

Result and discussion:

Providing more categorized information in discussion section based on diabetes criteria's (secondary outcomes) would be more interesting results for paper. For example, authors could explain effects of different dietary fats on fasting glucose/insulin, insulin resistance (HOMA-IR), HbA1c, etc... instead of only diabetes incidence (main outcome which is the most clinically relevant). This would add novelty to the meta-analysis.

The discussion does not support the different incidence of type 2 diabetes between Asia and the U.S population which is described in conclusion. Please discuss more in depth these results. 

Line 236: Did you consider the different types of trans fat in your analysis? Did the other paper consider the different transfat?

Limitations;

Regarding the different geographic location of reviewed papers, mediatory role of ethnic andgenetic could be considered as a limitation. 

Did any of the studies use biomarkers of fat intake to validate their FFQ or dietary recalls?

Conclusion:

Line 311: between dietary fats, fatty acids and type 2 diabetes incidence= please rewrite.

Line 317: too long sentence with many AND- please rewrite.

[LINK]

---

## [Decision Letter · Decision Letter 2]

30 Jul 2020

Dear Dr. Schlesinger,

Thank you very much for re-submitting your manuscript "Intake of dietary fats and fatty acids with the incidence of type 2 diabetes: a systematic review and dose-response meta-analysis of prospective observational studies" (PMEDICINE-D-20-00813R2) for review by PLOS Medicine.

I have discussed the paper with my colleagues and the academic editor and it was also seen again by original reviewers. I am pleased to say that provided the remaining editorial and production issues are dealt with we are planning to accept the paper for publication in the journal.

[LINK]

We look forward to receiving the revised manuscript by Aug 06 2020 11:59PM. 

Sincerely,

Clare Stone, PhD

Managing Editor 

PLOS Medicine

plosmedicine.org

Requests from Editors:

Title: please amend, needs to be "... and the incidence ..." 

Please include p values in the abstract and throughout for quantifiable data and where 95% Cis are given. 

Line 41 and some of the display items, comma rather than apostrophe in the publication number, please 

Line 46, we have "trend" which is often a trigger word for "non-significant association"; it looks like this is indeed a significant trend, however (please include the p value, I think at line 254)

- relatedly, at line 50 the "incidence decreased" refers to an association for alpha linolenic acid which is not obviously significant to my eye. Please include p values and state if significant or not. 

Abstract- "low to moderate” appears twice, please remove one of the mentions.

- in the abstract, the limitations sentence can be reorganized. The sentence on limitations should be explicit, startin with ‘limitations of this study are…..’ and please remove “Thus, the certainty of evidence is low to moderate” as this is too vague to be of value. 

Line 41 “In total, 15’070” should be “15, 070”

Line 57 - “Our results suggest weak associations” – there is either an association or not depending on p values. Please be clear. Weak association implies not and if so, don’t state there is one, please. 

Line 60- saying that plant fats prevent T2D would be over reaching. Please rethink some rerding, e.g. to substitution of plant for animal fats in diets

- to go back to p values, at line 264 we have "doses ... reduced T2D incidence" and then the estimate/95% CI look non-significant -please provide p value. Please do not make statements if they can’t be supported. 

Line 384, "... to our knowledge the first"

- reference 73 looks odd, also two or three others – please use referencing according to Vancouver style.

PRISMA attachment – please replace page numbers with sections and paragraphs as these change during formatting. 

Comments from Reviewers:

Reviewer #2: The authors have revised the manuscript in accordance with recommended changes; and answered questions previously raised.

[LINK]

---

## [Editor Report · Decision Letter 3]

10 Nov 2020

Dear Dr. Schlesinger, 

On behalf of my colleagues and the academic editor, Dr. Sanjay Basu, I am delighted to inform you that your manuscript entitled "Intake of dietary fats and fatty acids and the incidence of type 2 diabetes: a systematic review and dose-response meta-analysis of prospective observational studies" (PMEDICINE-D-20-00813R3) has been accepted for publication in PLOS Medicine. 

PRODUCTION PROCESS

Before publication you will see the copyedited word document (within 5 business days) and a PDF proof shortly after that. The copyeditor will be in touch shortly before sending you the copyedited Word document. We will make some revisions at copyediting stage to conform to our general style, and for clarification. When you receive this version you should check and revise it very carefully, including figures, tables, references, and supporting information, because corrections at the next stage (proofs) will be strictly limited to (1) errors in author names or affiliations, (2) errors of scientific fact that would cause misunderstandings to readers, and (3) printer's (introduced) errors. Please return the copyedited file within 2 business days in order to ensure timely delivery of the PDF proof. 

If you are likely to be away when either this document or the proof is sent, please ensure we have contact information of a second person, as we will need you to respond quickly at each point. Given the disruptions resulting from the ongoing COVID-19 pandemic, there may be delays in the production process. We apologise in advance for any inconvenience caused and will do our best to minimize impact as far as possible.

EARLY VERSION

PRESS

PROFILE INFORMATION

Thank you again for submitting the manuscript to PLOS Medicine. We look forward to publishing it. 

Best wishes, 

Richard Turner

Senior Editor 

PLOS Medicine

plosmedicine.org